# Mesoscale Dynamics and Transport in the North Brazil Current as revealed by the EUREC4A-OA experiment

Yan Barabinot<sup>1</sup>, Sabrina Speich<sup>1</sup>, Xavier Carton<sup>2</sup>, Pierre L'Hégaret<sup>3</sup>, Corentin Subirade<sup>4</sup>, Rémi Laxenaire<sup>5</sup>, and Johannes Karstensen<sup>6</sup>

**Correspondence:** Yan Barabinot (yan.barabinot@lmd.ipsl.fr)

Abstract. The North Brazil Current (NBC) rings are a key mechanism key features for interhemispheric water transport, facilitating the exchange between the South Atlantic Ocean and the North Atlantic South Atlantic and North Atlantic Ocean. However, significant uncertainties persist regarding the total volume transported by these structures and the properties of the water masses they advect. In this study, we integrate high-resolution in situ observations from the EUREC4A-OA field experiment with satellite altimetry to address these knowledge gaps. Using a novel methodology, we estimate that surface NBC rings transport approximately 1.5 Sv while subsurface eddies contribute between 0.4 Sv and 9.7 Sv underscoring their critical role in the regional total transport. Combined, these transports, may significantly contribute to closing the Atlantic Meridional Overturning Circulation transport at low latitudes. Surface NBC rings predominantly advect Salinity Maximum Waters and fresh waters from the Amazon River, whereas subsurface NBC rings play a critical role in transporting Eastern South Atlantic Central Waters, Western South Atlantic Central Waters, and Antarctic Intermediate Waters northward. We also found estimate that the heat transports transport by surface and subsurface NBC rings are here evaluated at is 5.8 TW and 0.3 which is much less than previous estimation TW, respectively, which is significantly lower than previous findings. Overall, these findings underscore the pivotal role of subsurface NBC rings as conduits for South Atlantic Waters across the equator and to the Tropical North Atlantic. This study confirms the intricate dynamics of NBC rings and their essential role into interhemispheric water transport.

## 1 Introduction

The North Brazil Current (NBC) (see in Figure 1) is a strong western boundary current that crosses the Equator and seasonally separates retroflects from the coast between 6° and 8°N to feed the North Equatorial Counter Current (Johns et al., 1990, 1998;

<sup>&</sup>lt;sup>1</sup>ENS, Université Paris Sciences et Lettres, Laboratoire de Météorologie Dynamique (LMD), 24 rue Lhomond, Paris 75005, France

<sup>&</sup>lt;sup>2</sup>Université de Bretagne Occidentale (UBO), Laboratoire d'Océanographie Physique et Spatiale (LOPS), IUEM, rue Dumont Durville, Plouzané 29280, France

<sup>&</sup>lt;sup>3</sup>Cedre, 715 rue Alain Colas, 29200 Brest, France

<sup>&</sup>lt;sup>4</sup>Université du Littoral Côte d'Opale (ULCO), 21 rue Saint-Louis, Boulogne-sur-Mer 62200, France

<sup>&</sup>lt;sup>5</sup>Université de la Réunion, Laboratoire de l'Atmosphère et des Cyclones (LACy), 15 avenue René Cassin - CS 92003, La Réunion, France

<sup>&</sup>lt;sup>6</sup>GEOMAR Helmholtz Centre for Ocean Research Kiel, Wischhofstraße 1-3, 24148 Kiel, Germany

Schott et al., 2002). This retroflection, as it is commonly called, is most developed from June to February and nearly absent from March to May (Johns et al., 1998). When the retroflection intensifies and forms current loops, large anticyclonic eddies, known as NBC rings, are shed into by the North Brazil Current.

**Figure 1.** Schematic of major ocean currents in the western tropical Atlantic superimposed on the regional bathymetry based on data from Smith and Sandwell (1997).

These rings have a mean <sup>1</sup> radius, defined as the location of maximum rotational flow amplitude, that can reach up to 200 km (Johns et al., 1990; Richardson et al., 1994; Fratantoni et al., 1995; Fratantoni and Richardson, 2006; Subirade et al., 2023). Their vertical extent, defined as the depth where velocity approaches zero, varies widely, from 200–300 m to as deep as 1000 m (Wilson et al., 2002; Fratantoni and Glickson, 2002; Johns et al., 2003; Fratantoni and Richardson, 2006). These large rings propagate northwestward along the South American coast with drift velocities ranging from 0.1 to 0.2 m.s<sup>-1</sup> (Ffield, 2005; Bueno et al., 2022; Subirade et al., 2023). While some rings are observed entering the Gulf of Mexico (Fratantoni and Richardson, 2006; Huang et al., 2021), the majority collide with the Lesser Antilles, breaking apart and dissipating through interactions with the topography (Fratantoni and Richardson, 2006; Jochumsen et al., 2010; Andrade-Canto and Beron-Vera, 2022). NBC rings exhibit a pronounced seasonal cycle in their characteristics. They tend to be weaker in spring, with smaller radii, lower amplitudes, shorter lifetimes, and slower azimuthal velocities. Additionally, they move faster but cover shorter distances during this season. Conversely, during summer, NBC rings are larger, stronger, and have a longer lifetime (Subirade et al., 2023).

<sup>&</sup>lt;sup>1</sup>mean in the sense of "averaged over angles"

In the literature, several values can be found for their Rossby number: between 0.13 and 0.26 according to Fratantoni et al. (1995), between 0.20 and 0.36 as shown by Richardson et al. (1994); Cruz-Gómez and Salcedo-Castro (2013), and up to 0.33 as estimated by Castelão and Johns (2011).

Estimates of the annual formation rate of NBC rings also vary across studies. Using color scanner imagery, Johns et al. (1990) found between 2 and 9 rings per year. Didden and Schott (1993), based on two and a half years of Geosat sea level anomaly data, estimated an average of 2.5 rings per year. Similarly, Richardson et al. (1994), using surface and subsurface drifters, found 2 rings per year. Altimetric studies provide a broader range: Goni and Johns (2001), analyzing Topex/Poseidon data from 1992 to 1998, reported 2 to 7 rings annually, and Goni and Johns (2003), using 10 years of data, estimated 3 to 7 rings per year. More recent studies, using various methods, generally place the annual formation rate between 4 and 8 rings (Sharma et al., 2009; Jochumsen et al., 2010; Mélice and Arnault, 2017; Aroucha et al., 2020; Subirade et al., 2023), with an average consensus of 4.5 rings per year.

45

Due to their large radii, NBC rings transport significant volumes of water. Using *in situ* data and the shallow water potential vorticity anomaly as a proxy for volume, Johns et al. (1990) and Fratantoni et al. (1995) estimated that each ring transports approximately 3 Sv (1 Sv =  $10^6 \text{m}^3$ ). This estimate was later corroborated by Didden and Schott (1993), using satellite data, and Richardson et al. (1994), employing floats. However, uncertainties introduced by limited data resolution were noted in these early studies (Fratantoni et al., 1995). Subsequently, Johns et al. (2003), using higher-resolution *in situ* data, proposed a reduced volume of 1.1 Sv per ring, a finding later supported by Bueno et al. (2022), who estimated 1.3 Sv. The improvement in data resolution appears to have refined these estimates downward. More recently, Subirade et al. (2023) calculated a significantly smaller transported volume of 0.12 Sv per ring, reigniting discussions on the true volume transported by NBC rings. Concerning the heat transport, Fratantoni et al. (1995) evaluated a total transport of 0.036 PW per ring while Garzoli et al. (2003) and Bueno et al. (2022) found an average of 0.07 PW. These variations highlight the sensitivity of heat transport estimates to methodology and spatial resolution.

NBC rings, by virtue of their large transported volumes, play a crucial role in the inter-hemispheric transport of mass, heat, salt, and various biogeochemical properties. They are particularly significant for the Atlantic Meridional Overturning Circulation (AMOC), as they help balance the southward export of North Atlantic Deep Water (NADW) with upper-ocean northward transport (Johns et al., 2003). To close the AMOC transport at low latitudes, Schmitz Jr and McCartney (1993) estimated that a canonical transport of 13 Sv is required. The contribution of NBC rings to this transport remains an open question. Furthermore, NBC rings facilitate the offshore advection and dispersal of fresh, nutrient-rich Amazon River waters, underscoring their ecological and physical significance (Johns et al., 1990; Fratantoni and Glickson, 2002; Reverdin et al., 2021; Olivier et al., 2024).

These observations primarily concern surface NBC rings, which are the most studied due to their prominent imprint on the ocean surface and their detectability in satellite data (Subirade et al., 2023). However, as identified by Johns et al. (2003); Fratantoni and Richardson (2006); Chen and Schiller (2024), NBC rings can be classified into three types based on their vertical structure: Surface NBC rings whose velocity field extends to -250 m; Subsurface Type I NBC rings that extend between -150 m and -700 m; Subsurface of Type II NBC rings, which extend between -200 m and -1000 m. Some studies have already

tempted to explain the vertical interaction between the surface and subsurface NBC rings (Napolitano et al., 2024). Although the formation of the surface type is well-known because the NBC retroflection can be detected by satellites, the generation and evolution of subsurface NBC rings of types I and II remains unclear. Johns et al. (2003) proposed two possible formation mechanisms for these subsurface rings: the deep retroflection of the NBC below the pycnocline or the interaction between the NBC and the Equatorial Undercurrent (EUC).

Subsurface NBC rings are not detectable via satellite data (Johns et al., 2003), which limits observations (Johns et al., 2003), limiting our understanding of their formation, inherent intrinsic dynamics, Rossby numbers, and transported volumes of water. Further research is needed associated water transport. This reflects a broader gap in our knowledge of the mesoscale dynamics in this region, which the present study aims to address. Improving this understanding is essential to elucidate these processes and quantify their role in contribution to ocean circulation.

In this context, the EUREC4A-OA field experiment was conducted to further characterize NBC rings and enhance our understanding of these critical ocean structures. EUREC4A-OA is part of the larger EUREC4A/ATOMIC initiative (Elucidating the Role of Clouds Circulation in Climate/Atlantic Tradewind Ocean–Atmosphere Mesoscale Interaction Campaign), which took place in January–February 2020 in the Western Tropical North Atlantic (WTNA) (Stevens et al., 2021). The aim of EUREC4A-OA was to study small-scale (0.1–100 km) ocean processes and their influence on air-sea fluxes.

The experiment involved four research vessels from Germany, French and the United States (Karstensen et al., 2020; Speich and Team, 2021; Quinn et al., 2021) as well as numerous uncrewed platforms (L'Hegaret et al., 2020; Stevens et al., 2021; L'Hégaret et al., 2023), which collectively provided extensive *in situ* measurements of the ocean and atmosphere. During the campaign, ocean eddies, including NBC rings, were identified and tracked over time. As surface NBC rings are visible from space, sampled eddies were tracked using the Tracking Ocean Eddies (TOEddies) automatic detection algorithm (Laxenaire et al., 2018, 2019, 2020, 2024; Ioannou et al., 2024) applied to Absolute Dynamic Topography (ADT) maps (Taburet et al., 2019). TOEddies was also used on Near Real Time ADT maps during the field campaign to guide the *in situ* sampling strategy (Speich and Team, 2021).

One of the EUREC4A-OA campaign's greatest strengths lies in the wide variety and high density of observing platforms used to sample mesoscale eddies. L'Hégaret et al. (2023) gathered data from the various devices used in the field campaign, applied a hierarchical quality control procedure and made the data interoperable. In particular, the data captured 15 cross sections that crossed mesoscale eddies. As a result, this study benefits from an unprecedented number of sampled sections, enabling detailed characterization of these structures and the ability to track some of them over time using *in situ* data. Notably, this includes subsurface eddies, which have not previously been examined in such detail.

The article is organized as follows: Section 2 describes the materials and methods, including the EUREC4A-OA experiment and the satellite data used for eddy detection. Section 3 outlines the methodology for detecting eddy boundaries and calculating their volumes. Section 4 presents the main results, focusing on mesoscale dynamics, eddy volumes, and transported water masses. Section 5 compares our findings with previous literature. Finally, the conclusions are summarized.

# 2 Materials and processing

## 2.1 EUREC4A-OA in situ data

The EUREC4A-OA field experiment was conducted in the Western North Tropical Atlantic during January–February 2020 as part of the EUREC4A-ATOMIC initiative. A comprehensive description of the ocean and atmospheric platforms deployed, along with the measurements collected during the experiment, is provided in Stevens et al. (2021); L'Hégaret et al. (2023).

To ensure interoperability of the data that was collected with the various platforms (ship, autonomous) L'Hégaret et al. (2023) developed a hierarchical data quality control procedures. The final data set comprises calibrated and cross-validated against quality-controlled CTD (Conductivity-Temperature-Depth) all vertical profiles measuring temperature, salinity, and velocity. Observations from different devices deployed along the same section were then concatenated and assigned uncertainty estimates.

This study focuses on sections sampled by the research vessels (R/Vs) L'Atalante (Speich and Team, 2021) and Maria S Merian (Karstensen et al., 2020). These sections include data from CTD, underway CTD (uCTD), and Moving Vessel Profiler (MVP) profiles. Upper-ocean velocity measurements were obtained using Ocean Surveyor Acoustic Doppler Current Profilers (ADCPs) onboard both ships. The R/V L'Atalante was equipped with 38 kHz and 150 kHz ADCPs, providing coverage from 20 m to below 1000 m depth, while the R/V Maria S Merian utilized 38 kHz and 75 kHz ADCPs, measuring from approximately 40 m to below 1000 m depth.

Additionally, two Argo floats (WMO nos. 6902966 and 6902957; http://doi.org/10.17882/42182), deployed in the core of a subsurface NBC ring, are included in the analysis.

# 120 2.2 Data processing

130

The reader is referred to Karstensen et al. (2020); Speich and Team (2021); Stevens et al. (2021) for further details of the data collection. This study uses the post-calibrated dataset presented in L'Hégaret et al. (2023).

During the EUREC4A-OA research cruises, data were often collected along vertical sections consisting of several vertical profiles. We define the resolution of a vertical section as the average distance between successive profiles along the same section. Since hydrographic and velocity measurements sampled the ocean at different resolutions, the two types of observations are treated separately. On average, the horizontal resolution of hydrographic data is 10 km, although in the best cases it can be as fine as 2.7 km. The vertical resolution of hydrographic data is 1 m. For velocity data, the horizontal resolution averages less than 0.3 km, while the vertical resolution is about 8 m (L'Hégaret et al., 2023). The first baroclinic Rossby radius of deformation in the equatorial region is approximately 150 km (Chelton et al., 1998). The horizontal resolution of the observing system is sufficient to resolve mesoscale eddies, including NBC rings.

To minimize noise, linear interpolation was used in both the horizontal and vertical directions. We define x the along cross section coordinate and z the depth coordinate. First order polynomial functions were chosen to avoid introducing artificial fields. The resulting interpolated grid has a typical horizontal spacing of 1 km and vertical spacing of 1 m (positive upward).

These values were chosen as a compromise between standardizing the data and maintaining the gradients. The data were then smoothed using a fourth-order numerical low-pass filter (implemented via scipy.signal.filt in Python).

The choice of cut-off scales is subjective and depends on the phenomena under investigation. For this study, which focuses on mesoscale eddies, we applied horizontal and vertical length scale cut-offs of  $L_x \ge 10$  km and  $L_z \ge 10$  m, respectively, to filter out submesoscale processes. The cut-off period was chosen to ensure that it exceeded the sampling resolution of the calibrated data.

# 140 2.3 Identification of eddies in *in situ* data collected by research vessels

In vertical hydrographic cross sections, eddies are identifiable by the vertical displacement of isopycnals, as their rotating flow primarily satisfies geostrophic equilibrium, with occasional cyclostrophic corrections (Cushman-Roisin and Merchant-Both, 1995; Penven et al., 2014; Ioannou et al., 2019). These displacements are often accompanied by changes in the sign of the velocity field orthogonal to the section in the horizontal direction. For accurate analysis of thermohaline anomalies in eddy cores, the ship transect must pass sufficiently close to the eddy center.

The position of the eddy center is estimated at a given depth using the method described in Nencioli et al. (2008). This approach employs a grid-based algorithm to identify the eddy center with high precision. The routine creates a rectangular grid around the ship transect. For each grid point, the velocity field measured along the transect is decomposed into radial and azimuthal components, assuming the grid point as the potential eddy center. The mean radial velocity component is then calculated. Each grid point is thus associated with an average radial velocity, and the true eddy center is identified as the point where this average radial velocity is minimized. This process is systematically repeated at each geopotential level, generating a two-dimensional map of potential vortex center positions across depth. Notably, this methodology was also employed during the experiment to predict the eddy center positions in real time, enabling the sampling strategy to target cross-sections as close as possible to the actual eddy centers. This dual application highlights its utility both for analysis and operational planning. It should be mentioned that this procedure can only be applied in case a complete eddy structure has been sampled. For instance, some cross-sections were carried out on half of an eddy structure (one part of the total velocity field).

## 2.4 Satellite data

To compare the surface signatures of the sampled eddies, we use satellite altimetry data combined with a detection algorithm based on Absolute Dynamic Topography (ADT) derived from these measurements.

The sampled eddies are identified and tracked over time using the TOEddies automatic detection algorithm (Laxenaire et al., 2018, 2019, 2020, 2024; Ioannou et al., 2024). During the field experiments, this detection was applied to ad-hoc Near Real Time (NRT) ADT maps. We used daily all-satellite sea surface height fields provided by the Copernicus Marine Service (https://marine.copernicus.eu/fr). This multi-satellite product integrates data from all available satellites at any given time and projects it onto a fixed grid with a resolution of 1/4°, covering the global ocean. The products used include the CNES-CLS18 MDT (Mulet et al., 2021), which serves as the standard for DUACS-DT2018 (Taburet et al., 2019).

The TOEddies method, developed based on the algorithm proposed by Chaigneau et al. (2009), has been employed in several studies investigating Atlantic Ocean dynamics. Examples include the origin and evolution of Agulhas Current rings (Laxenaire et al., 2018, 2019, 2020), the role of mesoscale eddies in meridional transport across the zonal South Atlantic GO-SHIP section during the MSM60 cruise (Manta et al., 2021), and mesoscale eddy dynamics in the EUREC4A-OA region (Subirade et al., 2023). It has also been used to analyze the effect of mesoscale eddies on the formation and transport of South Atlantic Subtropical Mode Water (Chen et al., 2022) and to develop a global mesoscale eddy atlas collocating sea surface detections with Argo float observations (Laxenaire et al., 2024).

Assuming that eddies are in geostrophic equilibrium, TOEddies identifies eddies as closed ADT contours containing a single local extremum. At any given time, the streamlines of an eddy correspond to the closed isolines of the daily ADT maps. The ADT, rather than the Sea Level Anomaly (SLA), represents the geostrophic stream function, as SLA is more sensitive to large Sea Surface Height (SSH) gradients linked to intense currents, quasi-stationary meanders, or eddies reflected in the Mean Dynamic Topography (MDT) (Pegliasco et al., 2021).

TOEddies detects local extrema (maxima and minima) in the ADT and identifies the outermost closed ADT contour surrounding each extremum. Additionally, the algorithm determines the contour where the mean azimuthal velocity is maximized, using geostrophic velocities derived from the ADT maps.

## 3 Methods

180

# 3.1 Dynamical variables

## Relative vorticity

To calculate the relative vorticity, derivatives in two different horizontal directions are needed. For a single section of a research cruise this is not possible without further assumptions. Following Halle and Pinkel (2003) method, we decompose the measured velocities into a cross-track component  $v_{\perp}$  and an along-track component  $v_{\parallel}$ . For a section crossing an eddy, we determine the location where  $|v_{\perp}|$  is minimum and consider this point to be a projection of the actual eddy center. The relative vorticity  $\zeta$  is then calculated using the following formula:

$$\zeta = \partial_r v_\perp + v_\perp / r,\tag{1}$$

190 where r is the radial distance from the location where  $|v_{\perp}|$  is minimum.

# **Ertel Potential Vorticity**

Here the 3D Ertel Potential Vorticity (EPV hereafter) formula (Ertel, 1942) is simplified and applied to *in situ* data collected in mesoscale eddies. Under the Boussinesq approximation and hydrostatic equilibrium, the vertical component of the linear momentum can be replaced by the hydrostatic approximation  $\partial_z p = -\rho g$ , where p is the pressure,  $\rho$  the total density and g the acceleration due to gravity. We also approximate  $1/\sigma$  by  $1/\sigma_0$  where  $\sigma$  refers to the potential density at atmospheric pressure

and  $\sigma_0$  as an average of  $\sigma$  on a vertical section. The EPV is calculated by

$$EPV = EPV_x + EPV_z = -\partial_z v_\perp \partial_r b + (\zeta + f_0) \partial_z b, \tag{2}$$

where  $f_0$  is the Coriolis parameter in the f-plane approximation and  $b = -g\sigma/\sigma_0$  the buoyancy. Defining the climatological mean for b, denoted as  $\overline{b}$ , the climatological mean of EPV, denoted as  $\overline{EPV}$  can be calculated according to  $\overline{EPV} = f_0 d\overline{b}/dz$ . We thus define the EPV anomaly by

$$\Delta EPV = EPV - \overline{EPV}.$$
(3)

This quantity is calculated on isopycnal surfaces. This quantity has been widely used to define the materially coherent core of eddies and is therefore of interest (Carton et al., 2010; Zhang et al., 2014; Barabinot et al., 2024).

Following the approach of Barabinot et al. (2024, 2025), we define the ratio between the anomaly of the vertical component,  $\Delta \text{EPV}_z = \text{EPV}_z - \overline{\text{EPV}}$  and the horizontal component  $\text{EPV}_x$  as  $\Delta \text{EPV}_z/\text{EPV}_x$ . It has been demonstrated that the eddy boundary is not locally defined but instead behaves as a frontal region influenced by submesoscale instabilities. These instabilities arise where the baroclinic term becomes comparable in magnitude to the vertical term (Hoskins and Bretherton, 1972; Hoskins, 1974; Buckingham et al., 2021a, b; Barabinot et al., 2024).

To capture the eddy core distinct from this turbulent frontal region, a criterion of the form:

$$210 \quad \frac{|\Delta EPV_z|}{|EPV_x|} > \beta, \tag{4}$$

where  $\beta \gg 1$  is employed. This criterion identifies the core water that remains relatively stable and isolated from turbulent mixing. Symmetric instabilities, which are prominent at the eddy boundary, can erode the eddy by modifying the properties of water parcels or generating small-scale turbulence (Armi et al., 1989; Haine and Marshall, 1998; D'Asaro et al., 2011; Thomas et al., 2016; Goldsworth et al., 2021). In contrast, the detected core water within the eddy is more stable and drifts along with the eddy without being significantly altered by the surrounding environment. As highlighted by Chen et al. (2022), this core may, if sufficiently homogeneous, contain what often is called Mode Waters.

In practice, the parameter  $\beta$  typically ranges from 10 to 50 (Barabinot et al., 2024, 2025), although variations in  $\beta$  have minimal impact on the calculated transported volume. This is because the gradient of the ratio  $\Delta \mathrm{EPV}_z/\mathrm{EPV}_x$  is very large at the eddy boundary (Barabinot et al., 2025).

#### 3.2 3D volume reconstruction

200

215

220

For an eddy volume reconstruction, we apply the methodology developed by Barabinot et al. (2025). On a 2D vertical section, eddy boundaries are identified using a conventional method, specifically a chosen isoline of  $\zeta$  (here set as  $\zeta = 0$ ). At a given depth, these boundaries are reduced to two points (indicated by the green dots in Figure 2). At this depth, the eddy center is computed using the routine developed by Nencioli et al. (2008) (denoted by the red dot in Figure 2).

The average eddy radius is then determined, and a small volume with element of height dz is computed, as illustrated in Figure 2. By summing all such contributions across the depth, 2. Summing these contributions over the vertical levels yields

Figure 2. A schematic representation of a ship transect (black squares line) sampling an eddy at a specific geopotential level, with velocity vectors represented by blue arrows. Using a predefined criterion and depth level, the eddy boundaries (green squares) are identified, while the eddy center (red square) is estimated following the method described by Nencioli et al. (2008). At this geopotential level, two radii  $R_1$  and  $R_2$  can be computed. An infinitesimal cylindrical volume is constructed using the average of  $R_1$  and  $R_2$ .

the total eddy volume  $\Omega$  is  $\Omega$ , given by

235

$$\Omega = \pi \sum_{n=1}^{N} \frac{(R_1^n + R_2^n)^2}{4} dz. \tag{5}$$

where  $R_1^n$  and  $R_2^n$  represent the two radii at the vertical level n (see in Figure 2). In the following, we define the average radius at level n as  $R^n = (R_1^n + R_2^n)/2$ . In this approach, we assume that the eddy is perfectly circular at each depth, although the radius and center may vary with depth.

Previous studies approximated NBC ring volumes using used a cylinder with a constant radius (Johns et al., 1990; Fratantoni et al., 1995; Johns et al., 2003; Bueno et al., 2022). In contrast, our method incorporates accounts for the potential baroclinic structure of NBC rings and take into account the potential eddy tilting(which means incorporates eddy tilting—that is, the variation in the location of the eddy centerwith depth) by use of the Nencioli et al. (2008) routine . eddy center's position with depth—by using the routine developed by Nencioli et al. (2008). It also accounts for depth-dependent variations in the eddy radius.

## 3.3 Heat content

245

Following Laxenaire et al. (2020), the Heat content (HC hereafter) transported by NBC rings is estimated by first calculating their HC anomaly. In the core of an eddy, HC is computed as:

$$HC = \int_{z_{inf}}^{z_{sup}} \int_{0}^{R(z)} \int_{0}^{2\pi} \rho(r, \theta, z) C_p \Theta(r, \theta, z) r dr d\theta dz,$$

$$(6)$$

where  $\rho = \sigma + 1000 \text{ kg.m}^{-3}$  is the density,  $C_p = 3991.87 \text{ J.kg}^{-1}.\text{K}^{-1}$  (according to the TEOS-10 standard) is the seawater specific heat capacity, and  $\Theta(r,z)$  [K] the conservative temperature, R(z) denotes the eddy radius at depth z while  $z_{sup}$  and  $z_{inf}$  are the upper and lower integration limits, corresponding to the vertical boundaries of the eddy.  $\theta$  is the angle in the cylindrical coordinate system. Using the reconstruction method described in Section 3.2, the HC can be approximated as:

$$HC = 2\pi \Delta z dz \sum_{n=1}^{N} \int_{0}^{R^{n}} \rho_{n}(r) C_{p} \Theta_{n}(r) r dr, \tag{7}$$

where  $\Delta z = z_{sup} - z_{inf}$  and the subscript *n* refers to the vertical level, as in Equation 5. The Heat Content Anomaly (HCA) is obtained by subtracting the local climatological heat content, denoted as  $\overline{\text{HC}}$ , from the HC computed within the eddy. To calculate the climatological variables, we used the methodology proposed by Laxenaire et al. (2019, 2020).

Climatological averages of temperature and salinity on geopotential levels are calculated using Argo float profiles collected over a 20-year period within a small area around the sampled eddy. These data are obtained from the Coriolis data center (dataselection.euro-argo.eu). A square with a side length of  $0.5^{\circ}$  is centered on the estimated eddy center, with the center located at the intersection of the diagonals. Denoting the climatological potential density and conservative temperature as  $\overline{\rho}$  and  $\overline{\Theta}$ , respectively, the  $\overline{HC}$  is given by:

255 
$$HC = 2\pi \int_{z_{inf}}^{z_{sup}} \int_{0}^{R(z)} \int_{0}^{2\pi} \overline{\rho}(z) C_p \overline{\Theta}(z) r dr dz.$$
 (8)

We then define the Heat Content Anomaly (HCA) as

$$HCA = \sum_{n=1}^{N} HC_n - \overline{HC_n}, \tag{9}$$

where  $\mathrm{HC}_n = 2\pi dz \int_0^{R^n} \rho_n(r) C_p \Theta_n(r) r dr$  is the heat content at depth level n and  $\overline{\mathrm{HC}_n}$  is the corresponding the climatological value.

#### 260 3.4 Water masses definition

To characterize the water masses advected by NBC rings, we rely on the established literature. Stramma and England (1999) identified the South Atlantic Central Water (SACW) as a dominant water mass in the upper tropical and subtropical Atlantic

Ocean. This water mass exhibits a nearly linear  $\Theta$ -S relationship (see also Sverdrup (1942)),  $\Theta$  and S being respectively the conservative temperature and absolute salinity. Mémery et al. (2000); Gordon (1981) refined this characterization by defining a potential density range of 25.6 kg.m<sup>-3</sup> to 26.5 kg.m<sup>-3</sup> for SACW, which is typically formed at the confluence of the Falkland/Malvinas and Brazil Currents..

In addition, Mémery et al. (2000) defined Salinity Maximum Water (SMW) as waters with potential density values below 25.6 kg.m<sup>-3</sup>, primarily formed due to excess evaporation in tropical regions. This water is also called Subtropical Underwater in the literature (Yu et al., 2018; Nie et al., 2020). More recent studies by Liu and Tanhua (2019, 2021) further categorized the South Atlantic Central Waters into two distinct subtypes: Eastern South Atlantic Central Water (ESACW) and Western South Atlantic Central Water (WSACW). In fact, this characterization is the recent version of the work performed by Poole and Tomczak (1999) who first analyzed the water mass structure in the Atlantic Ocean thermocline. ESACW is formed in the Agulhas retroflection region and has a conservative temperature between 9.44°C and 13.60°C, an absolute salinity between 34.9 g.kg<sup>-1</sup> and 35.40 g.kg<sup>-1</sup>, and a potential density range of 26.5 kg.m<sup>-3</sup> to 26.93 kg.m<sup>-3</sup>. WSACW forms near the South American coast between 30°S and 45°S. This water mass results from the mixing of three mode waters: SMW and Subtropical Mode Water (STMW) carried by the Brazil Current, and Subantarctic Mode Water (SAMW) transported by the Falkland/Malvinas Current (Álvarez et al., 2014). It shares many characteristics with the SACW defined by Stramma and England (1999) but has a narrower range of properties.

Finally, the Antarctic Intermediate Water (AAIW), characterized by a potential density range of 26.9 kg.m<sup>-3</sup> to 32.15 kg.m<sup>-3</sup>, occupies a deeper layer. Liu and Tanhua (2019, 2021) refined the definition by adjusting the lower boundary to 26.9 kg.m<sup>-3</sup>. However, regarding that according to the review of Xia et al. (2022), the Antarctic Intermediate Water can be identified by the salinity minimum when the water is introduced into the thermocline and is thus characterized by a potential density range of 27.0 kg.m<sup>-3</sup> to 27.2 kg.m<sup>-3</sup>.

To evaluate the origin of NBC ring-transported waters, we also examine North Atlantic Central Waters, specifically: Eastern North Atlantic Central Water (ENACW), which forms east of the Mid-Atlantic Ridge near  $20^{\circ}$ W and is characterized by conservative temperatures ranging from  $11.36^{\circ}$ C to  $13.82^{\circ}$ C, absolute salinities of  $35.69~\rm g.kg^{-1}$  to  $36.12~\rm g.kg^{-1}$ , and potential densities between  $26.89~\rm kg.m^{-3}$  and  $27.12~\rm kg.m^{-3}$ . Western North Atlantic Central Water (WNACW), formed north of the Lesser Antilles, between the Gulf Stream and the Mid-Atlantic Ridge, with conservative temperatures of  $17.51^{\circ}$ C to  $18.89^{\circ}$ C, absolute salinities of  $36.63~\rm g.kg^{-1}$  to  $36.82~\rm g.kg^{-1}$ , and potential densities of  $26.33~\rm kg.m^{-3}$  to  $26.55~\rm kg.m^{-3}$  (see also Liu and Tanhua, 2021; Ríos et al., 1992).

# 4 Results

290

265

# 4.1 Sampled eddies census in the North Brazil Current region

This section provides an overview of the dynamics observed by the two research vessels, R/V L'Atalante (referred to as AT) and R/V Maria S. Merian (referred to as MSM). For each observed eddy structure, we detail the numbers of cross-sections collected (as reported in L'Hégaret et al. (2023)) and analyze their temporal evolution using all available data.

In this study, we define surface eddies as those eddies with maximum velocity occurring at or just below the ocean surface. Subsurface eddies, on the other hand, are defined as those with maximum velocity located below the pycnocline. Key information on the sampled eddies is summarized in Table 1. Note that some eddies may appear on other sections. However, in some cases, as the cross-section has not crossed the full structure of the eddy, we cannot apply the Nencioli et al. (2008) routine. This table is useful to compare with other studies or satellites data. Name of eddies are defined afterwards.

Table 1. Dynamic properties of NBC $_{sub}1$ , NBC $_{sub}2$ , NBC $_{surf}1$  NBC $_{surf}2$ , and the cyclonic eddy.  $V_{max}$  denotes the maximum orthogonal velocity, and  $R_{max}$  represents the corresponding radius, both calculated using the Nencioli et al. (2008) method for determining the eddy centers. The locations of the eddy centers, identified by the same routine, are also reported. H is an order of magnitude of eddies vertical extension using maximum isopycnal deviations in eddies cores (sometimes density data are not available and we put a '-'). For NBC $_{surf}2$ , the eddy center is estimated using the TOEddies algorithm (indicated with a \*). Note that some eddies may appear on other sections. However, in some cases, as the cross-section has not crossed the full structure of the eddy, we cannot apply the Nencioli et al. (2008) routine.

| Name                          | $R_{max}$ [km] | $V_{max} [\mathrm{m.s}^{-1}]$ | H [m] | Location                                         | Cross-section |
|-------------------------------|----------------|-------------------------------|-------|--------------------------------------------------|---------------|
| NBC <sub>sub</sub> 1 (25/01)  | 91.9           | 0.99                          | -     | $(58.19^{\circ}W, 9.91^{\circ}N)$                | 4 AT          |
| NBC <sub>sub</sub> 1 (12/02)  | 70.0           | 0.96                          | 600   | $(58.10^{\circ}\text{W}, 10.10^{\circ}\text{N})$ | 32 AT         |
| NBC <sub>surf</sub> 1 (27/01) | 117            | 1.14                          | 100   | $(57.35^{\circ}W, 9.44^{\circ}N)$                | 5 AT          |
| NBC <sub>surf</sub> 1 (15/02) | 126            | 0.92                          | 80    | $(58.5^{\circ}W, 10.87^{\circ}N)$                | 28, 29 MSM    |
| NBC <sub>surf</sub> 2 (28/01) | 111*           | 0.78                          | 90    | $(51.49^{\circ}W, 7.28^{\circ}N)$                | 3 MSM         |
| NBC <sub>sub</sub> 2 (28/01)  | 142            | 0.83                          | 300   | $(51.97^{\circ}W, 8.70^{\circ}N)$                | 3 MSM         |
| NBC <sub>sub</sub> 2 (01/02)  | 109            | 0.79                          | -     | $(52.80^{\circ}W, 8.67^{\circ}N)$                | 13 AT         |
| Cyclone (26/01)               | 114            | 0.51                          | 100   | $(54.72^{\circ}W, 8.95^{\circ}N)$                | 3 MSM         |

# First double NBC eddies structure

300

305

The first remarkable structure observed during the campaign is illustrated in Figure 3. This structure features a vertical superposition of two NBC rings: one at the ocean surface, referred to as NBC<sub>surf</sub>1, and the other, an intra-thermocline eddy (previously described in the introduction as a subsurface type I eddy (Johns et al., 2003)), referred to as NBC<sub>sub</sub>1. NBC<sub>surf</sub>1 was sampled along cross-sections 5, 29, and 32 by the R/V L'Atalante and along cross-sections 28 and 29 of R/V Maria S. Merian. Panels (c) and (d) of Figure 4 show the vertical structure of NBC<sub>surf</sub>1 in cross-section 5 (AT) and 29 (MSM) using the dynamical Rossby number  $\zeta/f_0$  (Stegner and Dritschel, 2000). NBC<sub>sub</sub>1, on the other hand, was sampled along cross-sections 2, 3, 4, 5, 29, 32 by the R/V L'Atalante and along cross-sections 28 and 29 of the R/V Meria S. Merian. Panels (a) and (b) of Figure 4 provide the vertical structure of NBC<sub>sub</sub>1 in cross-section 4 and 32 (AT). These panels also show the presence of NBC<sub>surf</sub>1 at the ocean surface, positioned above the pycnocline.

Cross-sections 2, 3, 4 (AT) for NBC<sub>surf</sub> 1 and 5 (AT) for NBC<sub>sub</sub> 1 provide insights into the vertical structures from January 24 to 29. Using the methodology outlined in Section  $\frac{23}{2}$ , we reconstructed their 3D structure, revealing their relative positions

Figure 3. a): a) Three-dimensional reconstruction of  $NBC_{surf}1$  (green, January 27; blue, February 15) and  $NBC_{sub}1$  (orange, January 25; red, February 12), with eddy boundaries defined by the  $\zeta/f_0=0$  criterion. b): Top-down view of the same double-eddy structure. Subsurface eddy contours are represented by dashed lines, while surface eddy contours are shown as solid lines. Surface NBC eddies are modeled as idealized circles (with colored centers), whereas contours derived from altimetry exhibit less regular shapes (with dark centers). The regional bathymetry is represented by shaded colors, based on data from Smith and Sandwell (1997).

Figure 4. Normalized relative vorticity ( $\zeta/f_0$ ) of NBC<sub>sub</sub>1 for cross-sections 4 (panel (a)) and 32 (panel (b)), as observed by R/V L'Atalante. Corresponding  $\zeta/f_0$  for NBC<sub>surf</sub>1 is shown for cross-sections 5 (panel (c)) and 29 (panel (d)). Isopycnal surfaces are represented by dark lines. Note that in panel (a), density measurements do not extend beyond 300 m depth.

(Figure 3).  $NBC_{surf}1$  is represented in green, while  $NBC_{sub}1$  is shown in orange. Their estimated centers, calculated using the method of Nencioli et al. (2008), are marked with large dots in matching colors.

While NBC<sub>surf</sub>1 is detectable via the TOEddies algorithm, NBC<sub>sub</sub>1 remains invisible to satellite altimetry. Figure 3 confirms the effective detection of NBC<sub>surf</sub>1 using satellite data, consistent with Subirade et al. (2023). Altimetry is used here to validate our methodology by qualitatively comparing the positions of eddy centers. The Nencioli et al. (2008) method proves reliable, though a quantitative comparison of *in situ* and altimetric contours is not feasible due to differing boundary criteria ( $\zeta/f_0 = 0$  for *in situ* data versus the radius of maximum velocity for altimetry) and the inherent limitations of capturing true eddy shapes with ship transects.

A fortnight later, cross-sections 29, 32 from the R/V L'Atalante and cross-sections 28, 29 from the R/V Maria S. Merian provided insights into the vertical structure of NBC $_{surf}1$  and NBC $_{sub}1$  from February 11 to 16. In Figure 3, NBC $_{surf}1$  is represented in blue and NBC $_{sub}1$  in red. The maximum velocity contour of NBC $_{surf}1$  is depicted as a continuous blue line, with its center in a dark dot. A slight discrepancy is observed between satellite altimetry and *in situ* data in detecting NBC $_{surf}1$ . The difference in shape can be attributed to the use of  $\zeta/f_0=0$  as the boundary criterion, rather than the maximum velocity contour. Additionally, our methodology assumes eddy axisymmetry, which is not perfectly accurate in this case. Nevertheless, the eddy center estimated using the method of Nencioli et al. (2008) aligns well with the position derived from altimetry data. NBC $_{surf}1$  propagates northwestward toward the Lesser Antilles, whereas NBC $_{sub}1$ , being more influenced by topography, moves northeastward. Between January 28 and February 14, NBC $_{surf}1$  traveled a distance of 511 km in 28 days, while NBC $_{sub}1$  moved only 23.3 km. The estimated drifting velocity are  $2.1 \times 10^{-1}$  m.s $^{-1}$  for NBC $_{surf}1$  and  $9.6 \times 10^{-3}$  m.s $^{-1}$  for NBC $_{sub}1$ . While the drifting velocity of NBC $_{surf}1$  is consistent with the values reported in the literature (and even a bit higher according to Subirade et al. (2023) who found  $1.5 \times 10^{-1}$  m.s $^{-1}$ ), that of NBC $_{sub}1$  is significantly weaker. The size of NBC $_{surf}1$  remains quite constant over time. According to Figure 4, the dynamical signature of NBC $_{surf}1$  weakens over time, whereas that of NBC $_{sub}1$  maintains a consistent intensity and may even exhibit a slight decrease.

## Second double structure

The second notable double structure observed during the campaign is shown in Figure 5. This structure, like the first, consists of a vertical superposition of two NBC rings: a surface eddy NBC<sub>surf</sub>2 and a subsurface eddy NBC<sub>sub</sub>2. Panels (a) and (c) in Figure 6 highlight the superposition, although NBC<sub>surf</sub>2 is not fully sampled. NBC<sub>surf</sub>2 and NBC<sub>sub</sub>2 were observed along cross-sections 13 and 27 by the R/V L'Atalante and cross-sections 3, 5, and 8 by the R/V Maria S. Merian. While no cross-section fully captured NBC<sub>surf</sub>2, preventing the application of the Nencioli et al. (2008) method, NBC<sub>sub</sub>2 was fully sampled along cross-sections 3 and 8 (MSM) and 13 (AT). Using the methodology described in Section 2.3.2 and applying the criterion  $\zeta/f_0 = 0$  we reconstructed NBC<sub>sub</sub>2 in 3D, as shown in Figure 5. In this figure, NBC<sub>sub</sub>2 ris represented twice: in pink for January 27–28, 2020, and in purple for January 31, 2020. For NBC<sub>surf</sub>2, the TOEddies algorithm was used to plot its maximum velocity contour (continuous brown lines).

Cross-sections 3 and 5 (MSM) provide insights into the vertical structure of NBC $_{surf}$ 2 and NBC $_{sub}$ 2 from January 27 to 28, 2020. By January 31 (cross-section 13, AT), NBC $_{sub}$ 2 had progressed northwestward, with its 3D reconstruction, displayed in purple, illustrating this displacement. On February 6 (cross-section 27, AT), NBC $_{sub}$ 2's vertical structure was observed again, although the eddy center could not be estimated as the section-cross-section did not fully cross the core. NBC $_{sub}$ 2 traveled 91.4 km northwestward between January 28 and January 31, with a drifting velocity estimated at 0.26 m.s $^{-1}$ . During this interval, its maximum radius, determined using the criterion outlined in Section 43.1, contracted from 142 km to 113 km. In contrast, NBC $_{surf}$ 2 appears to remain stationary and shows reduced surface expression over time. This reduction may be attributed to interactions with the topography or possibly with the surrounding background flow, leading to crosionAlthough this feature appears as a closed SSH contour, it is difficult to determine whether it represents an eddy or a recirculation pattern.

Figure 5. a) Three-dimensional reconstruction of NBC<sub>sub</sub>2 in pink (January 28) and in purple (January 31), using  $\zeta/f_0 = 0$  as the boundary criterion. b) Top-down view of the double-eddy structure. Continuous lines show NBC<sub>surf</sub>2 detected by the TOEddies algorithm in light brown (January 28) and dark brown (January 31), while dashed lines represent reconstructed NBC<sub>sub</sub>2 (same color as panel a). The regional bathymetry is shaded (ETOPO2; Smith and Sandwell (1997)). Panel c) is similar as panel b) but we replaced the bathymetry with iso-ADT contours on day January 28 to show NBC<sub>surf</sub>2 still in the retroflection.

It may correspond to an eddy in the process of forming, but it remains attached to the current and thus does not exhibit the drift typically associated with isolated eddies.

355

The combined use of satellite altimetry (via the TOEddies algorithm) and *in situ* data allowed us to reconstruct the likely evolution of this double structure. The reader must take this scenario with care as we do not have enough data to validate it.

Figure 6.  $\zeta/f_0$  for NBC<sub>surf</sub>2 and NBC<sub>sub</sub>2 along a) cross-sections 3 (R/V Maria S. Merian) and b) cross-section 13 (R/V L'Atalante), with the corresponding orthogonal velocity  $v_{\perp}$  to the ship tracks. Isopycnals surfaces are shown as dark lines.

Figure 7 provides a comprehensive overview of its lifecycle. NBC<sub>sub</sub>2 and NBC<sub>surf</sub>2 likely formed together around  $7^{\circ}$ N, as suggested by cross-section 3 (MSM). This is consistent with the region of formation of subsurface NBC rings found in high resolution simulations (Napolitano et al., 2024). However, no definitive evidence from data supports this formation hypothesis. NBC<sub>sub</sub>2 subsequently propagated northwestward along the continental slope, whereas NBC<sub>surf</sub>2 remained quite stationnary near  $7^{\circ}$ N. It is probably trapped inside the retroflection as shown in Figure 5. Over time, the surface signature of NBC<sub>surf</sub>2 diminished, accompanied by a decrease in its radius. By February 1, NBC<sub>sub</sub>2 had migrated to approximately 8.67° but lacked a surface signal. On February 2, TOEddies detected the surface expression of NBC<sub>sub</sub>2 near 9°N. Cross-section 27 (AT) on

Figure 7. a) From January 25 to February 4,  $NBC_{surf}2$  progressively disappears in altimetry maps, as indicated by diminishing continuous contours, while  $NBC_{sub}2$  becomes visible in altimetry maps on February 2, represented by dashed contours. The colorbar for TOEddies contours and cross-sections dates are the same to indicate when cross-sections have been realized. Cross-section 13 (R/V L'Atalante, Figure 6) shows the presence of  $NBC_{sub}2$  without an ADT signature. Regional bathymetry is shaded (ETOPO2; Smith and Sandwell (1997)). b) Cross-section 27 (R/V L'Atalante) confirms the presence of  $NBC_{sub}2$  on February 6. c)  $NBC_{surf}2$  is clearly detected in ADCPs measurements along cross-section 5 (R/V Maria S. Merian) on January 29, but is absent in altimetry maps on February 4.

February 7 confirmed the presence of NBC $_{sub}$ 2 (panel b in Figure 7). Its maximum velocity was located at approximately 200 m depth, although its velocity field had now extended to the surface.

Both double structures exhibit similar characteristics: a surface eddy with a large radius and a velocity field confined above 250 m depth, coupled with a subsurface eddy featuring a smaller radius but a velocity field extending below 800 m depth. The interaction between the surface anticyclonic eddies and the intense, permanent pycnocline separating the two effectively masks the subsurface structures, making them challenging to detect in ADT maps.

# Cyclonic eddy

Cyclonic eddies in the North Brazil Current are hardly described in the literature. Fratantoni and Richardson (2006) showed that such eddies can emerge from the shear generated between two successive NBC rings. Figure 8 provides a clear example of this. The cyclonic shear between  $NBC_{surf}1$  and  $NBC_{surf}2$  at the ocean surface results in a localized reduction in ADT, inducing cyclonic water rotation (see panels (a) and (b) in Figure 8). The surface signature of this cyclonic eddy, evident in ADT maps (not shown here), diminishes over time as the surface radius of  $NBC_{surf}1$  decreased. The eddy disappears entirely from altimetry maps by February 11.

Cross-section 3 (MSM) provided a detailed view of the cyclonic eddy, as depicted in panels (a), (c), and (e) of Figure 8. The cyclonic velocity field is evident around x = 400 km, extending from the surface to z = -600 m, as shown in panel (e). Cross-section 6 (AT) also intersected the cyclonic structure, though only partially, as illustrated in panels (b), (d), and (f). The velocity field is relatively weak, with a maximum velocity of  $0.4 \text{ m.s}^{-1}$ , significantly lower than the velocities observed in NBC rings.

Notably, this cyclonic eddy may represent a superposition of two distinct cyclonic eddies: one located above the pycnocline and the other below. Panel (e) in Figure 8 clearly shows two distinct velocity maxima—one at the surface with a velocity of  $0.4 \text{ m.s}^{-1}$ , and another at z = -300 m with a velocity of  $0.3 \text{ m.s}^{-1}$ . An alternative hypothesis is that the pycnocline itself acts as a dividing layer, separating the cyclonic eddy into two parts: one confined to the ocean surface and the other situated below the pycnocline.

#### 4.2 NBC rings volume and transport estimates

To compute eddy volumes, it is essential to have both hydrographic and velocity data available on the same geopotential levels, along with a cross-section passing close to the eddy center at high resolution. Additionally, the cross-section must sample the entire structure of the eddy (rather than only one half) to apply the Nencioli et al. (2008) routine effectively. Since the volume calculation relies on several derivatives, the eddy signature—and particularly the gradients—must not be too small. These constraints make it challenging to compute volumes for certain cases. For instance, the cyclonic eddy cannot be analyzed due to its horizontal resolution exceeding 10 km and insufficient density gradients. Similarly, NBC $_{surf}$ 2 is excluded because a full cross-section of the eddy is unavailable, and NBC $_{sub}$ 2 is unsuitable due to weak horizontal gradients caused by coarse resolution. The reader can verify this by examining the deviation of isopycnals in panel (c) of Figure 6. This analysis underscores the critical importance of high-resolution spatial sampling, not only vertically but also horizontally, for thermohaline and velocity properties of eddies. Accurate volume calculations require resolving these properties with sufficient detail to capture the true structure of the eddy.

**Figure 8.** Cyclonic eddy sampled along cross-section 3 (R/V Maria S. Merian) and cross-section 6 (R/V L'Atalante). a-b) Absolute Dynamic Topography (ADT) maps displayed as shaded colors for January 26 and January 29, 2020, respectively. Eddy contours detected by the TOEddies algorithm are overlaid, with blue contours representing cyclonic eddies and red contours representing anticyclonic eddies. Ship cross-sections are superimposed in white for each date. c-d): Conservative temperature along cross-sections 3 and 6, respectively. e-f): Velocity orthogonal to the respective cross-sections.

Figure 9. Boundaries of  $NBC_{surf}1$  and  $NBC_{sub}1$  are shown in magenta in a and b, respectively, with  $\Delta EPV_z/EPV_x$  as the background field. c-d) Three-dimensional reconstructions using the methodology outlined in Section 3.2, with volume values indicated. Isopycnal surfaces are plotted as dark lines.

Using the methodology detailed in Section 3.2, we computed the volumes of NBC rings NBC $_{surf}1$  and NBC $_{sub}1$ . These rings were selected because their centers were precisely sampled by ship tracks (cross-section 5 (AT) for NBC $_{surf}1$ , cross-section 32 (AT) for NBC $_{sub}1$ ), minimizing side effects and reducing uncertainties in estimating eddy radii. The criterion used to calculate the transported volume is given by Equation 4. The results are presented in Figure 9.

As described in Section 3.1, eddy boundaries are identified by a local minimum of the  $|\Delta \text{EPV}z|/|\text{EPV}x|$  ratio (Figure 9, which correspond to turbulent regions influenced by sharp horizontal density gradients. Small-scale instabilities or lateral intrusions in these regions can modify the transported water mass (Barabinot et al., 2024; Armi et al., 1989; Joyce, 1977, 1984; Ruddick et al., 2010).

Moving toward the core, the ratio increases sharply before stabilizing, indicating a clear boundary for the eddy. For this analysis, eddy boundaries are determined using  $|\Delta \text{EPV}z|/|\text{EPV}x| = 30$ . Contrary to previous studies suggesting cylindrical shapes, the NBC rings analyzed here exhibit a top-shaped structure. NBC<sub>surf</sub>1 has a volume of  $1.63 \times 10^{12} \text{ m}^3$ , while NBC<sub>sub</sub>1 has a volume of  $5.2 \times 10^{12} \text{ m}^3$ . To compare our results with those obtained using earlier methodologies that assume eddies are perfect cylinders (Fratantoni et al., 1995; Johns et al., 2003; Bueno et al., 2022), we compute the volume using the formula  $\pi R_{max}^2 H$ , where  $R_{max}$  represents the radius of maximum velocity (provided in Table 1). The vertical extent of the eddy core, H, is estimated by examining its structure. Panels (a) and (b) in Figure 9 suggest  $H \approx 150 \text{ m}$  for NBC<sub>surf</sub>1 and  $H \approx 500 \text{ m}$  for NBC<sub>sub</sub>1. Using this approach, the computed volume for NBC<sub>surf</sub>1 is  $6.45 \times 10^{12} \text{ m}^3$ , while that for NBC<sub>sub</sub>1 is  $7.69 \times 10^{12} \text{ m}^3$ . Assuming eddies as purely barotropic structures clearly leads to an overestimation of the transported volume. The difference between our methodology and the cylindrical assumption is more pronounced for the surface structure than for the subsurface structure, as the latter is inherently more cylindrical in shape.

The transported volume of an eddy with total volume  $\Omega$ , drifting velocity  $V_d$  and maximum diameter  $d_m$ , can be approximated as  $V_d\Omega/d_m$ . Here,  $d_m/V_d$  represents the time it takes for an eddy center to travel a distance  $d_m$ , assuming a straight trajectory. Using drifting velocities of  $2.1 \times 10^{-1} \text{ m.s}^{-1}$  for NBC $_{surf}1$  and  $9.6 \times 10^{-3} \text{ m.s}^{-1}$  for NBC $_{sub}1$ , the transported volumes are estimated at 1.5 Sv and 0.35 Sv respectively (averaged over 28 days). This indicates that the transport contribution of NBC $_{surf}1$  is approximately 4.5 times higher than that of NBC $_{sub}1$ . However, as shown in Figure 3, NBC $_{sub}1$  may be constrained by the continental slope, which significantly reduces its drifting velocity. If the drifting velocity of NBC $_{sub}2$  (0.26 m.s<sup>-1</sup>) is used as a reference for NBC $_{sub}1$ , the transported volume increases substantially, reaching 9.7 Sv, demonstrating the potential significance of subsurface eddies under different conditions.

#### 4.3 Water masses advected by NBC rings

435

NBC $_{surf}1$  and NBC $_{surf}2$  primarily transport SMW, as evidenced by their positions in the  $\Theta$ -S diagrams and their alignment with SMW characteristics. In contrast, NBC $_{sub}1$  and NBC $_{sub}2$  exhibit a composition dominated by ESACW, WSACW, and AAIW. ESACW is prevalent in the eddy core, WSACW forms the upper boundary, and AAIW defines the lower boundary. Notably, none of the analyzed NBC rings transport North Atlantic water masses.

Panel (c) in Figure 11 emphasizes the escape of water masses from the core of NBC<sub>sub</sub>2 along isopycnal surfaces. The cyclonic eddy sampled by R/V Meria S. Merian near  $x=400~\rm km$  captures some of these escaping waters. This is evidenced by green and orange patches visible in the panel, located in the subsurface region of the cyclonic eddy around  $z=-200~\rm m$ . Additionally, the  $\Delta \rm EPV_z/EPV_x=30$  boundary encloses waters that remain trapped within the eddy core, underscoring its role in isolating certain water masses.

Figure 10. Analysis of water masses advected by  $NBC_{surf}1$  and  $NBC_{sub}1$ , comparing eddy cores (blue) with surrounding background waters (red). SACW = South Atlantic Central Waters (yellow), ESACW = Eastern South Atlantic Central Waters (orange), WSACW = Western South Atlantic Central Waters (green), AAIW = Antarctic Intermediate Waters (purple), ENSACW = Eastern North Atlantic Central Waters (blue), WNSACW = Western South Atlantic Central Waters (pink). a–b) Total water masses in cross-sections 5 and 32 (R/V L'Atalante), respectively. c–d) Trapped waters within  $NBC_{surf}1$  and  $NBC_{sub}1$ , respectively. e–f) Surrounding background waters in cross-sections 5 and 32 (R/V L'Atalante), respectively.

Figure 11. Distribution of water masses within the core of sampled eddies. a)  $NBC_{surf}1$ , b)  $NBC_{sub}1$ , c)  $NBC_{surf}2$ , and d)  $NBC_{sub}2$ . Isopycnals surfaces are indicated by dark lines. Contours of  $NBC_{surf}1$  and  $NBC_{sub}1$ , defined by  $\Delta EPV_z/EPV_x=30$ , are highlighted in magenta.

Further analysis of two Argo floats (N°6902966 and 6902957) deployed purposefully in NBC $_{sub}$ 2 during the EUREC4A-OA field campaign, provides insights into the temporal evolution of hydrographic properties within the core of NBC $_{sub}$ 2. Figures 12 and 13 illustrate their respective trajectories and time series. Both floats were launched at the same time. The fact they share an identical trajectory emphasizes the material coherence of the eddy. The floats are estimated to have exited the eddy core, as indicated by the cessation of looping patterns, around April 1. In addition to analyzing water masses, we also computed temperature anomalies on isopycnal surfaces relative to climatological averages (Barabinot et al., 2024, 2025), with the methodology for climatological averages detailed in Section 3.3. These anomalies reveal significant temperature

Figure 12. Trajectories of Argo floats N $^{\circ}$ 6902966 and 6902957 within the core of NBC<sub>sub</sub>2 over time, illustrating the material coherence of the eddy. Both floats follow a similar looping pattern until exiting the core of the eddy around April 1. Regional bathymetry is shaded (ETOPO2; Smith and Sandwell (1997)).

variations along isopycnal surfaces, enabling the identification of water masses at specific density levels. As shown in Figure 13, a pronounced negative temperature anomaly is observed between February 2 and March 14, within the isopycnal surfaces 26.5 kg.m<sup>-3</sup> and 27 kg.m<sup>-3</sup>. This anomaly corresponds to ESACW and WSACW trapped within the core of NBC<sub>sub</sub>2. After March 14 the negative anomaly decreases as the floats appear to leave the eddy core. However, it remains uncertain whether the trapped water masses are fully released from the core or continue to circulate within the eddy. These Figures figures highlight the link between material coherence, which is a consequence of closed trajectories, and the thermohaline coherencethat is that is, the trapped water does not have the same properties of the surrounding waters (Barabinot et al., 2025).

## 4.4 Heat transport

455

Using the methodology described in section Section 3.3, we computed the heat content (HC) and heat content anomaly (HCA) for NBC $_{surf}1$  and NBC $_{sub}1$ . The results are summarized in Table 2, highlighting that the heat content anomaly represents only a small fraction of the total heat content. Our methodology to compute eddies volume requires both hydrological and velocity data on the same geopotential level. Sometimes, either one or the other is missing on vertical cross-sections especially for eddies NBC $_{surf}2$  and NBC $_{sub}2$ , that is why results are only shown for NBC $_{surf}1$  and NBC $_{sub}1$ .

**Figure 13.** Panels (a) and (b): Time series showing water mass composition and temperature anomalies observed by Argo floats N° 6902966 (a) and 6902957 (b). The *x*-axis represents time in days, while the *y*-axis indicates depth in meters. Isopycnal surfaces are depicted with dark lines. The time series captures significant hydrographic variations within NBC<sub>sub</sub>2, including a pronounced negative temperature anomaly in panels (c) and (d) associated with Eastern and Western South Atlantic Central Waters (ESACW and WSACW).

Figure 14 illustrates the depth-dependent heat content anomaly for  $NBC_{surf}1$  and  $NBC_{sub}1$ . For the surface anticyclonic eddy  $NBC_{surf}1$ , the HCA remains positive at all depths. This pattern arises because the downward-decreasing anticyclonic rotation depresses isopycnal surfaces, allowing the eddy to retain warmer waters compared to the ambient ocean.

Conversely, the HCA for the subsurface anticyclonic eddy  $NBC_{sub}1$  exhibits a more complex behavior, with both positive and negative values depending on depth. This is because the vertical spacing of isopycnal and isotherm surfaces increases due to the anticyclonic rotation, resulting in colder waters than the environment in the upper part of the eddy (above its median

Figure 14. Heat Content Anomaly (HCA) [J] as a function of geopotential depth for  $NBC_{surf}1$  (left) and  $NBC_{sub}1$  (right). Positive values indicate warmer water relative to the surrounding ocean, while negative values indicate colder water.

plane) and warmer waters in the lower part of the eddy (below its median plane). As  $NBC_{sub}1$  lacks vertical symmetry about its median plane (see Figures 9 and 11), the integration over the entire depth results in a net positive total HCA.

**Table 2.** Heat Content (HC) and Heat Content Anomaly (HCA) for NBC<sub>surf</sub>1 and NBC<sub>sub</sub>1, along with their respective heat transports.

| Name                  | HC [J]                 | HCA [J]                | HC transported [PW] | HCA transported [TW] |
|-----------------------|------------------------|------------------------|---------------------|----------------------|
| NBC <sub>surf</sub> 1 | $2.090 \times 10^{21}$ | $6.48 \times 10^{18}$  | 1.88                | 5.82                 |
| NBC <sub>sub</sub> 1  | $6.002 \times 10^{21}$ | $3.988 \times 10^{18}$ | 0.41                | 0.27                 |

# 465 5 Discussion

# 5.1 Eddy dynamics

During the EUREC4A-OA field campaign, four NBC rings were identified along the North Brazil Current (NBC) pathway. In two instances, a surface ring was observed above a subsurface ring situated below the pycnocline. These findings confirm the two types of NBC rings previously reported in the literature: surface NBC rings and subsurface NBC rings of type I (Johns et al., 2003; Fratantoni and Richardson, 2006). However, no evidence of subsurface NBC rings of type II, as described by Johns et al. (2003), was found during this field study.

As illustrated in Figure 6, the surface and subsurface structures likely formed together through the retroflection of the NBC, further supporting the existence of vertical coupling between these two types of eddies (Napolitano et al., 2024). These two types of eddies do not share the same maximum velocity, nor the same core, and we therefore consider them to be distinct structures. Figure 7 demonstrates that surface NBC rings often mask the presence of subsurface structures until their collapse. Interestingly, subsurface eddies are sufficiently strong to leave an imprint on the sea surface, which raises an important consideration for satellite-based analyses: surface signatures may represent subsurface structures as it is the case for Mediterranean water eddies or Meddies (Ciani et al., 2017; Bashmachnikov and Carton, 2012). Consequently, relying solely on surface characteristics could lead to inaccurate representations of the true eddy core properties. The observation of two subsurface NBC rings within a 28-day period highlights their relatively frequent occurrence, suggesting that these structures should not be considered exceptions.

In their article, Napolitano et al. (2024) studied the coupling and splitting between surface and subsurface NBC rings using a high resolution simulations and theoretical arguments. They showed that the coupling helps both surface and subsurface NBC rings to pass the continental bump near  $54^{\circ}$ W and  $8^{\circ}$ N (visible in our Figure 12 and in their Figure 8). In their simulation, surface NBC rings are indeed less affected by the topography and are able to drag subsurface eddies. As a result, in non-coupling cases, authors often observed situations where surface NBC rings continue on their northeastward trajectory, while subsurface eddies remain trapped in the topography, particularly at the location of the small continental slope bump. In this article, we observe the exact opposite in Figure 7 where NBC<sub>surf</sub>2 collapses near the small bump (probably not because of topography but because of the retroflexion) and NBC<sub>sub</sub>2 drifts northwestward. This is also shown in the video of the supplementary materials. In Napolitano et al. (2024), authors also proved that surface NBC rings were crucial for the propagation and lifetime of subsurface NBC rings. Here, as shown in Figure 7, we interestingly found that a subsurface NBC ring can drift alone towards the Lesser Antilles without being dependent on a surface NBC ring. These differences emphasize the complexity of the North Brazil Current mesoscale dynamics.

Another key finding is the presence of NBC-cyclone-NBC-nlbC-cyclone-NBC-ring systems, which enhance the northwest-ward transport of water. These vortex aggregates, formed by the interaction between NBC rings and cyclones, contribute to the redistribution of Amazon freshwater and South Atlantic water masses. As illustrated in Figures 8 and 11, although cyclones are much weaker than NBC rings, they are capable of trapping water masses. In the specific case studied here, the transported volume associated with the cyclones appears to be minimal. However, this finding underscores the need for future studies to consider NBC rings as part of a horizontally coupled system rather than isolated structures. Such an approach could provide a more comprehensive understanding of their role in ocean circulation and the transport of water masses.

# 5.2 Transport

Using a refined methodology that accounts for the actual vertical extent of NBC rings, we estimate that surface NBC rings transport a volume of approximately 1.5 Sv. This value is consistent with previous studies, although slightly higher than earlier estimates (Johns et al., 2003; Bueno et al., 2022). The discrepancy may arise from differences in methodologies, particularly the inclusion of the full vertical structure of the NBC rings in our analysis. By comparison, subsurface NBC rings transport

an estimated volume going from 0.35 Sv up to 9.7 Sv. This transport is not negligible relative to that of surface NBC rings. However, due to the lack of precise generation rate or drifting velocity estimates for subsurface rings, it remains challenging to quantify their overall contribution to the total northwestward transported volume. The value of 9.7 Sv could lead to overturn the transport paradigm in this region, which tends to be estimated via surface eddies only. The question on the importance of subsurface eddies remain open. However, according to these results, their contribution is not negligible.

Subsurface NBC rings play a crucial role in the interhemispheric exchange of water masses. As illustrated in Figure 10,11, and 13 subsurface NBC rings are responsible for the northward advection of key water masses originating in the South Atlantic, including waters from the Agulhas leakage, the Falkland/Malvinas Current, and the Southern Ocean. In contrast, surface NBC rings only transport primarily transport Salinity Maximum Waters (SMW) and freshwater inputs from the Amazon. While the volumetric transport by subsurface NBC rings has to be confirmed, their role as a conduit for South Atlantic water masses into the North Atlantic is significant, underscoring their importance in connecting these two basins.

Regarding heat transport (see Table 2), surface NBC rings are more efficient than subsurface rings due the greater transport associated with surface rings compared to subsurface ones is attributed both to their higher temperatures and the warmer waters they carry. Consequently, the total heat transported by surface structures is greater to the larger translation velocities of surface eddies. The heat transports by surface and subsurface NBC rings are here evaluated at 5.82 TW and 0.27 TW which is much less than previous estimation. Indeed, Fratantoni et al. (1995) found 35 TW per ring while Garzoli et al. (2003) and Bueno et al. (2022) found 70 TW per ring. This can be inferred to the difference of methodology. Indeed, in previous studies, authors used bulk formulas for the heat content and volume transport estimation. The transported volume was estimated assimilating NBC rings to cylinders and the heat transport was computed with the bulk formula  $\rho C_p \Omega \Delta T$  with  $\rho$  assumed as the constant density in the layer,  $\Omega$  the volume of the cylinder and  $\Delta T$  the difference of temperature between the core of the eddy and the surrounding waters, which is different from our formula 6. Moreover, in previous studies, it was determined that the temperature anomaly  $\Delta T$  could reach 15 °C, whereas here we estimate the temperature anomaly to be no more than 4 °C for both NBC rings (calculated on geopotential levels relatively to the climatological mean). With our values and a generation rate of 4.5 rings per year, the NBC rings contribution to the meridional heat transport at low latitudes, estimated at 1 PW by Ganachaud and Wunsch (2000), drops to 2.7% whereas previous studies found between 20% to 50%. The question remains open.

## **5.3** Relation to the AMOC

The southward export of North Atlantic Deep Water (NADW) within the Atlantic Meridional Overturning Circulation (AMOC) must be balanced by a compensatory northward transport in the upper ocean. Schmitz Jr and McCartney (1993) estimated that a canonical northward volume transport of approximately 13 Sv is required to close the AMOC at low latitudes. Based on the volume transport estimates derived in this study -1.5 Sv for surface NBC rings and 0.35 Sv for subsurface NBC rings—combined with an average NBC ring generation rate of  $4.5 \pm 1.1$  rings per year from the literature, we can assess the contribution of NBC rings to this canonical transport. This generation rate mostly corresponds to the generation of surface rings. Therefore, we extrapolate here for subsurface rings.

During the one-month observation period, two surface and two subsurface eddies were generated. Assuming an equal generation rate for surface and subsurface NBC rings, the total annual volume transported by NBC rings is estimated to be approximately  $8.3 \pm 2.1$  Sv. This value is consistent with the findings of Johns et al. (2003), who reported a northward transport of 9 Sv by NBC rings. However, as shown in Section 5.2, the transport of a single subsurface eddy can reach up to 9.7 Sv. This suggests that the contribution of just two subsurface NBC rings could exceed the canonical 13 Sv required to close the AMOC. This discrepancy leaves the question open for further investigation.

These findings underscore the significant role of NBC rings in the interhemispheric exchange of mass and heat within the AMOC system. While our results suggest that NBC rings alone could theoretically account for the full 13 Sv needed to close the AMOC, uncertainties persist regarding the specific contribution of subsurface NBC rings. Future research should prioritize determining accurate generation rates, volumes and transports for these structures to better quantify their role in the AMOC.

#### 6 Conclusions

560

565

This study leveraged on the unique dataset of high-resolution *in situ* observations from the EUREC4A-OA field experiment, complemented by satellite altimetry, to investigate the mesoscale dynamics and transport within the North Brazil Current (NBC) pathway between January 20 and February 20, 2020. The *in situ* measurements, characterized by their unparalleled spatial and temporal resolution, provided detailed insights into the structure and evolution of mesoscale and submesoscale processes, making this dataset uniquely suited to studying the NBC system. Our analysis revealed that NBC rings were frequently present in pairs, comprising one surface and one subsurface structure. These paired rings are often vertically coupled but may decouple due to interactions with the continental slope. Additionally, cyclonic eddies were observed to form in the shear zones between successive NBC rings, although their contribution to water transport is considerably smaller than that of NBC rings.

Using a novel volume computation methodology, we estimated that surface NBC rings transport approximately 1.5 Sv, while subsurface NBC rings exhibit a transported volume ranging from 0.35 Sv to as much as 9.7 Sv. Surface NBC rings primarily advect Salinity Maximum Waters and freshwater from the Amazon River, whereas subsurface NBC rings transport key water masses, including Eastern South Atlantic Central Waters, Western South Atlantic Central Waters, Antarctic Intermediate Waters and respective subtropical and subpolar Mode Water. Subsurface NBC rings are pivotal in connecting the South Atlantic with the Northern Hemisphere, facilitating the interhemispheric transfer of South Atlantic Waters across the equator Equator. The substantial transport capacity of subsurface NBC rings challenges the existing paradigm, which previously emphasized surface NBC rings as the dominant drivers of water transport in this region. However, uncertainties regarding the generation rates, volumes, transports and drifting velocities of subsurface NBC rings remain and must be resolved to fully validate this conclusion.

This study underscores the intricate dynamics and critical role of the NBC system in interhemispheric water mass exchange, offering new insights into its contribution to the global ocean circulation and climate systems. Future research should prioritize resolving uncertainties surrounding subsurface NBC rings and exploring their broader implications for oceanic and climatic processes.

Data availability. The ADT produced by Ssalto/Duacs distributed by CMEMS (Mulet et al., 2021; Taburet et al., 2019), accessed on January, 19, 2021: https://resources.marine.copernicus.eu

The concatenated R/Vs Atalante and Maria S Merian hydrographic and velocity data (L'Hégaret et al., 2023; Speich and Team, 2021; 575 Karstensen et al., 2020) are freely available on the SEANOE website: https://www.seanoe.org/data/00809/92071/, accessed on 15 March 2021.

The TOEddies (Laxenaire et al., 2024) atlas is available on the SEANOE website: https://www.seanoe.org/data/00917/102877/

The Argo float data (Argo, 2025) are available from the Coriolis Global Data Assembly Centre (GDAC): https://www.coriolis.eu.org/Data-Products/Data-selection

Author contributions. SS, XC, JK and RL conceived and conducted the field campaign as well as its objectives in terms of eddy sampling and assessments. YB performed the main diagnostics and YB and SS conceived the original draft. Additions and revisions have been carried out by RL, JK, XC, PLH and CS

Competing interests. The authors report no conflict of interest.

Acknowledgements. This research was supported by the European Union's Horizon2020 research and innovation program under grant agreement no. 817578 (TRIATLAS; https://triatlas.w.uib.no) and agreement no. 101060452 (OCEAN:ICE, https://ocean-ice.eu) that has been funded together with the UK Research and Innovation, the Centre National d'Etudes Spatiales through the TOEddies and EUREC4A-OA projects, the French National Program LEFE INSU, IFREMER, the French Vessel Research Fleet, the DATA TERRA French Research Infrastructures AERIS and ODATIS, IPSL, the Chaire Chanel Program of the ENS Geosciences Department, and the EUREC4A-OA JPI Ocean and Climate Program. We thank all the people who collected, processed and made public the data as well as all the institutions for which these people worked in particular GEOMAR Helmoltz Centre for Ocean Research Kiel. We also warmly thank every captain and crew of the RVs Atalante and Maria S. Merian without whom this study could not have been carried out. Yan Barabinot is supported by a Ph.D. grant from the Ecole Normale Supérieure de Saclay. Xavier Carton acknowledges support from UBO and a CNES contract EUREC4A-OA.

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
