# Peer review of "Mesoscale Dynamics and Transport in the North Brazil Current as revealed by the EUREC4A-OA experiment"

_EGUsphere, 2025_

## Referee Comment (RC1)

**Mesoscale Dynamics and Transport in the North Brazil Current as revealed by the EUREC4A-OA experiment**

**1   General Comments**

This paper investigated the dynamics North Brazil Current (NBC) rings and computed volume and heat transports of the surface and subsurface NBC rings. Their computations relied on the recent in situ observations from the EUREC4A-OA field experiment and satellite altimetry. Compared to previous studies, their computations utilized the vertical structure of the eddies provided by the unparalleled vertical and spatial resolution of the EUREC4A-OA field experiment. Previous studies emphasized surface rings as the dominant force of water transport in this region. They estimated that the subsurface brazil current rings transport water mass between 0.4 Sv and 9.7 Sv, and the surface rings transport about 1.5 Sv. Their estimates cast doubt on previous estimates, but their estimates of subsurface ring transport still has large uncertainty. The drift velocity and transported volume can depend on factors like surrounding flow, topography etc, which I think the paper should remind the reader of those factors. Heat transport of the surface and subsurface rings were estimated to be lower than previous estimates. I find Methods and assumptions are clearly explained, and their computations support their conclusions.

**2   Specific Comments**

The followings are my detailed questions and suggestions.

1. The paragraph starting from line 344 does not have enough data to support the statements. Is it appropriate to include the paragraph?

2. The section on Cyclonic eddy seems to be isolated in the paper. I don't see the connection of this section with the rest of the paper.

3. In section 4.2, eddy boundaries are determined using $|\Delta EPV_z|/|EPV_x|$. However, section 3.2 line 215 says eddy boundaries are identified using a chosen isoline of $\zeta$. I get confused which method is used for the calculations in section 4.2 and what are the purposes of the two methods of identifying eddy boundaries.

4. On line 397, eddy boundaries are determined using $|\Delta EPV_z|/|EPV_x| = 30$. How sensitive is the volume estimate to this criteria?

5. The drifting velocity of $NBC_{sub}2$ is used to estimate transported volume. What about also using the drift velocity of $NBC_{surf}1$ to make a lower estimate of transported volume of surface rings? It seems that the transported volume of both surface and subsurface rings can have large variability due to factors like background flow, topography, seasonality etc.

**3 Technical Corrections**

1. In Figure 12 panels (c) and (d), the x-axis extents are different from panel (a) and (b). I think it will be better to have the same x-axis limits if you have the data.

2. In figure 9, it seems that WNACW and ENACW are shaded, but not mentioned in the caption.

3. Line 303 and 332 mention "Section 1", but it's not clear which section they are referring. Similarly, "Section 4" on line 341 is also confusing.

4. Line 225 should delete word "used".

5. In equation (7), $\Delta z$ is the layer depth. I find it confusing it to express it as $z_{sup} - z_{inf}$, which is the depth of the whole eddy.

6. Add space on line 421 between "km" and "captures".

7. Add space on line 440 between "section" and "3.3".

---

## Author Comment (AC1)

**Reply referee 1**

We would like to thank the reviewer for his/her review and constructive feedback. We appreciate the effort and time the reviewer has invested in evaluating our work. Please find our point-to-point response below in blue.

**General Comments**

*This paper investigated the dynamics North Brazil Current (NBC) rings and computed volume and heat transports of the surface and subsurface NBC rings. Their computations relied on the recent in situ observations from the EUREC4A-OA field experiment and satellite altimetry. Compared to previous studies, their computations utilized the vertical structure of the eddies provided by the unparalleled vertical and spatial resolution of the EUREC4A-OA field experiment. Previous studies emphasized surface rings as the dominant force of water transport in this region. They estimated that the subsurface brazil current rings transport water mass between 0.4 Sv and 9.7 Sv, and the surface rings transport about 1.5 Sv. Their estimates cast doubt on previous estimates, but their estimates of subsurface ring transport still has large uncertainty. The drift velocity and transported volume can depend on factors like surrounding flow, topography etc, which I think the paper should remind the reader of those factors.*

We thank the reviewer for his/her constructive comment. Indeed, both the drift velocity and the transported volumes may be influenced by factors such as the surrounding flow and underlying topography. We have highlighted these aspects in the discussion section of the revised manuscript..

*Heat transport of the surface and subsurface rings were estimated to be lower than previous estimates. I find Methods and assumptions are clearly explained, and their computations support their conclusions.*

**Specific Comments**

*1. The paragraph starting from line 344 does not have enough data to support the statements. Is it appropriate to include the paragraph?*

We thank the reviewer for bringing this to our attention. The paragraph starting from line 344 deals with the evolution of the second vertical superimposition of two NBC rings. We agree that we do not have enough data to discuss the formation of this vertical superimposition of two NBC rings, nor its evolution after February 7. However, the evolution from January 25 and February 7 is supported by Figure 4 and 6, which shows the surface signature of this structure as well as cross-sections carried out in the structure. This paragraph, Figure 4 and 6 show that (1) a surface NBC ring, that is detected by the TOEddies algorithm can remain trapped in the retroflection, (2) a subsurface NBC ring can have a surface signature (this is also supported by Figure 4. c.), (3) a subsurface NBC ring is not necessarily trapped within the retroflection region and can drift away independently of its surface counterpart. Although more precise temporal data are not available, we are confident in these observations, which

are supported by both in situ measurements and altimetry data. We therefore consider it important to retain this paragraph, as it offers valuable insight into the surface and subsurface dynamics of the region.

*2. The section on Cyclonic eddy seems to be isolated in the paper. I don't see the connection of this section with the rest of the paper.*

We thank the reviewer for his/her constructive comment. One of the objectives of this study was to construct an eddy census based on in situ data. We agree that cyclonic eddies are not central to our analysis and are generally less significant in the transport of water masses in this region. However, previous studies have shown that some cyclonic eddies can be intense and capable of transporting water masses toward the French West Indies (Fratantoni et al. 2006). As NBC rings are, by definition, anticyclonic eddies, cyclonic eddies in the region are often under-documented. Here, we contribute a description of the vertical structure of a cyclonic eddy, an aspect that remains relatively rare in the literature.

*3. In section 4.2, eddy boundaries are determined using $|\Delta EPV_z|/|EPV_x|$. However, section 3.2 line 215 says eddy boundaries are identified using a chosen isoline of $\zeta$. I get confused which method is used for the calculations in section 4.2 and what are the purposes of the two methods of identifying eddy boundaries.*

We thank the reviewer for his/her constructive comment. We agree that this choice may appear confusing at first glance, but several factors motivated it. First, in Section 3.2, our aim was to compare eddies contours identified by the TOEddies algorithm with in situ data (Figure 2.b. and Figure 4.b). To do so, we selected a criterion based solely on the velocity field. Second, the resolution of the hydrographic data did not allow for the computation of $|\Delta EPV_z|/|EPV_x|$ across all cross-sections. In some cases, the spacing between uCTD or CTD profiles was too large to reliably estimate the required gradients. To ensure consistency throughout Section 3.2., we therefore used relative vorticity $\zeta$ in Figures 2 and 4. Moreover, using $\zeta$ provides insight into the dynamical interactions between the surface and the subsurface NBC rings, which have been shown to be significant in previous studies (e.g., Napolitano et al. 2024).

*4. On line 397, eddy boundaries are determined using $|\Delta EPV_z|/|EPV_x| = 30$. How sensitive is the volume estimate to this criteria?*

We thank the reviewer for his/her important comment. We acknowledge that this is not a commonly used criterion. It was first proposed by Barabinot et al. (2024) to estimate the volume of mesoscale eddies while accounting for their turbulent boundaries. The method was further developed in Barabinot et al. (2025), where the authors discuss the appropriate threshold values to be used.We kindly refer the reviewer to these studies for a detailed justification of the approach. Below, we provide a summary of the main idea, illustrated with an example.

Figure 1 displays the quantity $|\Delta EPV_z|/|EPV_x|$ for the subsurface NBC ring sampled along cross-section 32 of the RV Atalante. In the figure, the core of the eddy is saturated in dark red, indicating values of $|\Delta EPV_z|/|EPV_x| > 50$. A sharp gradient

can be observed at the eddy boundary, for example, at z=-300 m, the ratio increases from 1 to 50 over a horizontal distance of 2.8 km.

Using a threshold of $|\Delta EPV_z|/|EPV_x|$ = 50 instead of 30 reduces the estimated eddy volume by only 2.3%. The rationale is that $|EPV_x|$ should be negligible compared to $|\Delta EPV_z|$, ensuring that the effect of submesoscale instabilities near the eddy edge can be safely neglected.

Barabinot et al. (2024, 2025) concluded that the eddy volume derived from this isoline-based method is not very sensitive to the specific threshold value chosen.

[Figure]

Figure 1. $|\Delta EPV_z|/|EPV_x|$ for the subsurface NBC ring sampled by the cross-section 32 of the Atalante.

*5. The drifting velocity of $NBC_{sub}2$ is used to estimate transported volume. What about also using the drift velocity of $NBC_{surf}1$ to make a lower estimate of transported volume of surface rings? It seems that the transported volume of both surface and subsurface rings can have large variability due to factors like background flow, topography, seasonality etc.*

We thank the reviewer for his/her helpful comment. The drift velocity of $NBC_{surf}1$ was used to estimate the transported volume of surface rings as described between lines 407

and 414. In contrast $NBC_{surf}2$ exhibits no drift as it remains trapped within the retroflection region.

We agree that the transported volume of both surface and subsurface rings can vary significantly due to factors such as background flow, topography, seasonality etc. Our estimates are specific for the conditions observed in January and February. We have added a sentence in the discussion section to clarify this point.

*Technical Corrections*

*1. In Figure 12 panels (c) and (d), the x-axis extents are different from panel (a) and (b). I think it will be better to have the same x-axis limits if you have the data.*

We thank the reviewer for bringing this to our attention. The axes are indeed the same; however, the x-axes of panels (b) and (c) are shlightly compressed due to the presence of the colorbars. In the revised version of the manuscript, we adjusted the figure accordingly.

*2. In figure 9, it seems that WNACW and ENACW are shaded, but not mentioned in the caption.*

We thank the reviewer for bringing this to our attention. We agree, and the abbreviations WNACW and ENACW have been added to the figure caption in the revised manuscript.

*3. Line 303 and 332 mention "Section 1", but it's not clear which section they are referring. Similarly, "Section 4" on line 341 is also confusing.*

We thank the reviewer for bringing this to our attention. We have modified the numbers in the revised version of the manuscript.

*4. Line 225 should delete word "used".*

We thank the reviewer for bringing this to our attention. The typographical error has been corrected in the revised manuscript..

*5. In equation (7), $\Delta z$ is the layer depth. I find it confusing it to express it as $z_{sup} - z_{inf}$, which is the depth of the whole eddy.*

We thank the reviewer for bringing this to our attention. The typographical error has been corrected in the revised manuscript.

*6. Add space on line 421 between "km" and "captures".*

We thank the reviewer for bringing this to our attention. The typographical error has been corrected in the revised manuscript.

*7. Add space on line 440 between "section" and "3.3".*

We thank the reviewer for bringing this to our attention. The typographical error has been corrected in the revised manuscript.

References:

- Fratantoni, D. M., & Richardson, P. L. (2006). The evolution and demise of North Brazil Current rings. Journal of Physical Oceanography, 36(7), 1241-1264.
- Napolitano, D. C., Carton, X., & Gula, J. (2024). Vertical interaction between NBC rings and its implications for South Atlantic Water export. Journal of Geophysical Research: Oceans, 129(4), e2023JC020741.
- Barabinot, Y., Speich, S., & Carton, X. (2024). Defining mesoscale eddies boundaries from in‑situ data and a theoretical framework. Journal of Geophysical Research: Oceans, 129(2), e2023JC020422.
- Barabinot, Y., Speich, S., & Carton, X. (2025). Assessing the thermohaline coherence of mesoscale eddies as described from in situ data. Ocean Science, 21(1), 151-179.

---

## Author Comment (AC2)

**Reply referee 2**

We would like to thank the reviewer for his/her review and constructive feedback. We appreciate the effort and time the reviewer has invested in evaluating our work. Please find our point-to-point response below in blue.

*The manuscript entitled "Mesoscale Dynamics and Transport in the North Brazil Current as revealed by the EUREC4A-OA experiment" by Barabinot et al. uses a series of in situ observations (CTD, uCTD, ADCP, MVP, Argo floats) and satellite measurements to characterize the 3-D structure of surface and subsurface North Brazil Current rings (size, Rossby number, depth, T-S properties) and estimate the associated mass and heat transport that plays an important role in the interhemispheric water exchange and, consequently, to the AMOC. The manuscript is very well written and exposed logically; the results are clear and important, particularly when it comes to quantifying subsurface eddy transport once these eddies are not detectable from altimetry. Also, the authors have done a great job with the literature review and putting their results into context by comparing them with previous studies. I am suggesting some (minor) additional work to improve the clarity of the manuscript. Thus, I recommend the publication of this manuscript after minor revisions.*

*I suggest adding a figure (Figure 1) of the region of interest with some of the key currents and the NBC rings for broader context (either a schematic or satellite ADT map). This will help the readers to (1) visualize the region of interest and the dynamics associated and (2) put your results into a broader context in terms of interhemispheric water exchange and link to the AMOC.*

We thank the reviewer for his/her relevant suggestion. We agree that including a figure of the study region with the major currents would enhance the clarity of the manuscript. Accordingly, we have added such a figure in the revised version.

*I. 1. NBC rings are not a mechanism but features. I suggest writing 'The North Brazil Current (NBC) rings are key features…'.*

We thank the reviewer for bringing this to our attention. We agree and have corrected the wording in the revised version of the manuscript.

*I.2. Better written as '…South Atlantic and North Atlantic Ocean'.*

We thank the reviewer for bringing this to our attention. The sentence has been corrected in the revised version of the manuscript.

*I.3. Water masses are associated with T-S properties, so to me, the 'properties of water masses' doesn't make much sense. I suggest writing '…by these structures and the water masses they advect.'*

We thank the reviewer for bringing this to our attention. The sentence has been corrected in the revised version of the manuscript.

*l. 10 is a bit confusing. Suggestion: 'We estimate that the heat transport by surface and subsurface NBC rings is 5.8 TW and 0.3 TW, respectively, which is significantly lower than previous findings.'*

We thank the reviewer for bringing this to our attention. In the revised manuscript, we have replaced our original sentence with the reviewer's suggested wording.

*l. 12. And -> to? 'for South Atlantic Waters across the equator to the Tropical North Atlantic.'*

We thank the reviewer for his/her suggestion. In the revised manuscript, we have replaced our original sentence with the reviewer's suggested wordingion.

*l. 16-19. 'This retroflection' -> it is not clear that there is a retroflection based on the first line. Also, NBC rings are \*formed\* by NBC shedding; they do not shed \*into\* NBC. I suggest rewriting the first paragraph. Adding Figure 1 (broad scope) would help as well.*

We thank the reviewer for bringing this to our attention. In the revised version, we have rewritten the paragraph following the reviewer's suggestion. The addition of Figure 1 will help clarify the context.

*l. 73. 'Further research is needed…' this sentence makes it sound that the current manuscript doesn't cover this, which is not the case. I suggest rephrasing to indicate that it's a gap and/or that you are addressing this in the current manuscript.*

We thank the reviewer for his/her suggestion. In the revised manuscript, we have replaced our original sentence with the reviewer's suggested wordingion.

*l. 81. Typo in the L'Hegaret et al. 2020 citation.*

We thank the reviewer for bringing this to our attention. The typographical error has been corrected accordingly in the revised manuscript.

*l. 122 and l. 131. Add somewhere in this section that the first baroclinic Rossby radius of deformation in the equatorial region is ~ >150 km, which means that the horizontal resolution of the instruments is high enough to resolve mesoscale eddies/NBC rings (assuming you need 4 to 6 grid points to solve a feature).*

We thank the reviewer for his/her relevant suggestion. Following Chelton et al. (1998), we have added the order of magnitude of the baroclinic Rossby radius of deformation in the revised manuscript.

*l. 137. Make it clear that the changes in sign of the velocity happen in the horizontal direction*

We thank the reviewer for his/her comment. We have clarified the sentence in the revised manuscript.

*l. 220. Across -> along depth*

We thank the reviewer for bringing this to our attention. We have  corrected the wording  in the revised version of the manuscript.

*l. 225 Typo. Remove 'used'.*

We thank the reviewer for bringing this to our attention. The typographical error has been corrected accordingly in the revised manuscript.

*l. 228. I suggest adding at the end: '…, as well as the depth variation in the eddy radius.'*

We thank the reviewer for his/her relevant suggestion. The suggestion has been added in the revised version of the manuscript.

*l. 233. Seawater specific heat capacity*

We thank the reviewer for bringing this to our attention. The wording has been corrected in the revised manuscript.

*l. 272. 'regarding that the' -> 'according to the'*

We thank the reviewer for bringing this to our attention. The wording has been corrected in the revised version of the manuscript.

*Figure 2 and Figure 4 are very nice!*

We thank the reviewer for his/her comment.

*l. 299-300. What makes the authors believe that the surface and subsurface eddies are not part of the same eddy structure? Is the small tilting enough to claim that these are individual eddies? Just curious, but I think including this in the text is relevant.*

We thank the reviewer for his/her relevant question. We agree that this is not an obvious statement; however, several arguments support the conclusion that the surface and subsurface eddies do not belong to the same eddy structure.

First, the velocity field and the stratification observed along cross section 5 of the RV Atalante indicate that the pycnocline (defined as the depth of the maximum Brunt Vaisala frequency) divides the vertical structure of the flow into two distincts layers, each with its own velocity maximum. The figure below provides a zoomed view of this vertical separation: the surface structure is visible from 0 to 300 km at depths shallower than -150 m depth, while the subsurface structure appears from 0 to 150 km at depths greater than -150 m. We observe that the positive velocity core of the surface structure lies directly above the negative velocity core of the subsurface structure.

[Figure]

Second, If we define eddies on their Ertel PV anomalies, the two structures exhibit distinct anomaly values and are separated by the pycnocline. The figure below shows the Ertel PV and its anomaly along cross-section 5 of the RV Atalante.

[Figure]

Third, using the criterion proposed by Barabinot et al. (2024) – namely, the ratio ΔEPVz /EPVx – to define the cores of NBC rings, we show that the surface and subsurface structures do not share the same core (see Figure 9 of the article).

This observation holds for the duration of the campaign, although the available data are insufficient to track the full life cycles of the two eddies. It remains possible that they were part of a single structure prior to the observations.

*Figure 10. Great figure! Very instructive.*

We thank the reviewer for his/her comment.

*l. 333 typo '.' After NBC_sub2*

We thank the reviewer for bringing this to our attention. The typographical error has been corrected accordingly in the revised manuscript.

*Figure 4. What is being called NBC_surf2 eddy is actually a recirculation feature. Although this feature shows as a closed SSH contour and would probably be detected in other eddy methods, this is not strictly speaking an eddy. It is just a recirculation within the retroflection (or an about-to-form eddy, but there it is still attached to the current, thus it will not drift away as an eddy would). I suggest discussing this.*

We thank the reviewer for his/her relevant remark. This question also arose during the preparation of the manuscript. We considered the feature to be a vortex in the process of forming, but as its signature weakens over time, it is difficult to draw a definitive conclusion. We have added the discussion suggested by the reviewer in the revised manuscript.

*l. 501 Typo. remove 'only transport'.*

We thank the reviewer for bringing this to our attention.The typographical error has been corrected accordingly in the revised manuscript.

*l. 505 The large transport in the surface rings compared to the subsurface ones is due to the higher temperatures but also to the larger translation velocities associated with the surface eddies.*

We thank the reviewer for bringing this to our attention. We have added this argument to support our conclusions.

*l. 421 space missing between 'km' and 'capures'.*

We thank the reviewer for bringing this to our attention. The typographical error has been corrected accordingly in the revised manuscript.

References:

- Barabinot, Y., Speich, S., & Carton, X. (2024). Defining mesoscale eddies boundaries from in‑situ data and a theoretical framework. Journal of Geophysical Research: Oceans, 129(2), e2023JC020422.
- Chelton, D. B., DeSzoeke, R. A., Schlax, M. G., El Naggar, K., & Siwertz, N. (1998). Geographical variability of the first baroclinic Rossby radius of deformation. Journal of Physical Oceanography, 28(3), 433-460

---

## Author Response (AR2)

**Corrections - referee 1**

Please find our point-to-point response below. We thank the reviewer again for his/her comments.

*"The authors have addressed most of my questions. I just have two minor comments.*

*1. I made a typo in the previous review comments. In the last paragraph of section 4.2, the drifting velocity of $NBC_{sub}2$ is used to estimate transported volume of $NBC_{sub}1$. I suggest using the drift velocity of $NBC_{surf}2$ to make a lower estimate of transported volume of surface rings, because $NBC_{surf}2$ has a lower drift velocity that $NBC_{surf}1$.*

We thank the reviewer for his/her remark and suggestion. A sentence has been added in line 419 of the revised manuscript.

*2. In figure 10, the authors mentioned the shading "WNACW" and "ENACW" in the caption now. However, in the caption they use "WNSACW" AND "ENSACW". I suggest they use consistent acronyms."*

We thank the reviewer for bringing this to our attention. The caption has been corrected accordingly.